# Utilization of Lead Slag as In Situ Iron Source for Arsenic Removal by Forming Iron Arsenate

**DOI:** 10.3390/ma15217471

**Published:** 2022-10-25

**Authors:** Pan Chen, Yuxin Zhao, Jun Yao, Jianyu Zhu, Jian Cao

**Affiliations:** 1School of Minerals Processing and Bioengineering, Central South University, Changsha 410083, China; 2Key Laboratory of Hunan Province for Clean and Efficient Utilization of Strategic Calcium-Containing Mineral Resources, Central South University, Changsha 410083, China; 3School of Water Resource and Environment Engineering, China University of Geosciences (Beijing), Beijing 100083, China; 4Key Laboratory of Biohydrometallurgy of Ministry of Education, Central South University, Changsha 410083, China

**Keywords:** arsenic wastewater, lead slag, iron arsenate, scorodite

## Abstract

In situ treatment of acidic arsenic-containing wastewater from the non-ferrous metal smelting industry has been a great challenge for cleaner production in smelters. Scorodite and iron arsenate have been proved to be good arsenic-fixing minerals; thus, we used lead slag as an iron source to remove arsenic from wastewater by forming iron arsenate and scorodite. As the main contaminant in wastewater, As(III) was oxidized to As(V) by H_2_O_2_, which was further mineralized to low-crystalline iron arsenate by Fe(III) and Fe(II) released by lead slag (in situ generated). The calcium ions released from the dissolved lead slag combined with sulfate to form well-crystallized gypsum, which co-precipitated with iron arsenate and provided attachment sites for iron arsenate. In addition, a silicate colloid was generated from dissolved silicate minerals wrapped around the As-bearing precipitate particles, which reduced the arsenic-leaching toxicity. A 99.95% removal efficiency of arsenic with initial concentration of 6500 mg/L was reached when the solid–liquid ratio was 1:10 and after 12 h of reaction at room temperature. Moreover, the leaching toxicity of As-bearing precipitate was 3.36 mg/L (As) and 2.93 mg/L (Pb), lower than the leaching threshold (5 mg/L). This work can promote the joint treatment of slag and wastewater in smelters, which is conducive to the long-term development of resource utilization and clean production.

## 1. Introduction

The anthropogenic activities associated with massive enterprises like mining and metallurgical operations result in large amounts of strongly acidic wastewater (pH < 2) with significant concentrations of arsenic and sulfuric acid [1,2,3]. Large amounts of As_2_O_3_ fumes produced during the smelting process enter wastewater during the flue-gas scrubbing acid-generation step [4,5]. As a result, arsenic in the wastewater primarily takes the form of H_3_AsO_3_ [6,7], which is far more hazardous and difficult to remove than pentavalent arsenic. Although much research has been done in the field of wastewater treatment by adsorption [8,9], ion exchange [10], biological treatment [11], and membrane filtration [12], the most widely used method for arsenic-containing wastewater is still the neutralization precipitation method [13], in which arsenic is eliminated as calcium arsenate (Ca_3_(AsO_4_)_2_), calcium arsenite (Ca_3_(AsO_3_)_2_), and arsenic sulfide (As_2_S_3_). The effective and harmless treatment of acidic arsenic-containing wastewater has received extensive attention in recent years, and targeted research work has also been carried out [14]. According to research by Kong et al. [15], using UV light to accelerate the release of hydrogen sulfide from thiosulfate resulted in 99.9% of the arsenic being removed, and there was no hydrogen sulfide (H_2_S) pollution. To prevent the creation of hazardous waste, they devised a UV/formic acid (UV/HCOOH) technique for the reductive recovery of arsenic from extremely acidic wastewater in the form of a monolithic arsenic (As (0)) product [16]. These studies have shown positive findings for the removal of arsenic, but due to cost and process constraints, large-scale applications are difficult to implement. Studies have also been done on the elimination of arsenic by hydrothermal scorodite (FeAsO_4_.2H_2_O) synthesis using in situ iron sources like magnetite [17], limonite [18], and hematite [19,20]. Scorodite is regarded as the optimal arsenic-fixing mineral [21] due to its high arsenic-loading capacity (20–30%) and low solubility in extremely acidic conditions [22]. Synthetic scorodite is often generated in an acidic solution and high temperature conditions [23,24]. However, it has been demonstrated that lower reaction temperatures can also encourage the generation of scorodite under adequate pH and iron supersaturation circumstances [25]. As the scorodite method of arsenic removal necessitates the introduction of a significant amount of iron sources, the development of the non-ferrous smelting industry can be better supported if effective arsenic removal efficiency can be attained using bulk industrial solid waste. As a by-product of lead smelting, lead slag is often disposed of as solid waste due to its low utilization value. However, its high iron content and strong alkalinity make it acceptable for use as an in situ iron source for the treatment of acidic arsenic-containing wastewater. Li et al. [26] used LZSS (Lead–Zinc smelting slag) as an in situ Fe donor and neutralizer to remove arsenic from wastewater in the form of scorodite under 90 °C, and a 98.42% removal efficiency of arsenic was achieved with an initial As concentration of 7530 mg/L and H_2_SO_4_ concentration of 53,420 mg/L. In addition, we successfully obtained 99.95% As removal efficiency at room temperature with an initial As concentration of 6500 mg/L and H_2_SO_4_ concentration of 56,000 mg/L. The As-bearing precipitate exhibited stable leaching characteristics with an As concentration of 3.36 mg/L and Pb concentration of 2.93 mg/L due to semi-encapsulation by ferric hydroxide and silicate colloid.

Therefore, we suggested using lead smelting slag as an in situ iron source and neutralizer in the wastewater treatment process for acidic arsenic-containing wastewater. On the basis of previous research, the effects of oxidant dosage, solid-to-liquid ratio, temperature, and reaction time on the precipitate’s properties of arsenic removal efficiency were investigated. Meanwhile, the mechanism of arsenic removal and fixation by co-precipitation, precipitation, adsorption, and semi-encapsulation of lead slag at room temperature was established by analyzing the phase structure, chemical composition, valence conversion, and morphological transformation of the As-bearing precipitate using X-ray diffraction (XRD, Bruker Corp., Karlsruhe, Germany), Fourier transform infrared spectroscopy (FTIR, Thermo Fisher Scientific, Waltham, MA, USA), scanning electron microscopy–energy dispersive spectroscopy (SEM-EDS, TESCAN MIRA LMS, Brno, Czech Republic), X-ray photoelectron spectroscopy (XPS, Thermo Fisher Scientific K-Alpha, Waltham, MA, USA), and transmission electron microscopy–energy dispersive spectroscopy (TEM-EDS, Thermo Fisher Scientific, Talos F200X, Waltham, MA, USA).

## 2. Methods and Materials

### 2.1. Materials Characterization

The lead slag and smelting wastewater used in the study was collected from a lead smelting plant in Gejiu, Yunnan Province, China. Before the experiment, the lead slag was taken for air-drying and ground to pass through a 75μm sieve. The mineral constituents of the lead slag were determined by X-ray powder diffraction (XRD) using a pressed sample ground to < 75μm. Chemical composition of the lead slag was analyzed by X-ray fluorescence (XRF, PANalytical Axios, Almelo, The Netherlands) as shown in Table 1. The concentration of the main component of the arsenic wastewater was measured by inductively coupled plasma–optical emission spectrometer (ICP-OES), sulfuric acid concentration was titrated with NaOH [6], and the water sample was diluted 100 times and measured three times to take the average value and calculate the standard deviation, as shown in Table 2. The chemical agents used in the experiment, including H_2_O_2_ (AR, 30 wt.%, Aladdin, Shanghai, China), H_2_SO_4_ (AR, Huihong, Changsha, China), and NaOH (≥96%, Macklin, Shanghai, China), were laboratory-grade and all solutions were prepared with deionized water (DI) at standard atmospheric pressure and room temperature.

### 2.2. Experiment Procedures and Methods

Batch leaching experiments were carried out to investigate the main influencing factors of arsenic removal efficiency by lead slag, including oxidant addition dosage, solid–liquid ratio, reaction time, and temperature. The lead slag and arsenic-containing wastewater were mixed in a conical flask, with oxidant dosage ranging from 0–10%, solid–liquid ratio ranging from 1:15–1:3 g/mL, and reaction temperature ranging from 25 °C to 85 °C. All batch experiments were conducted in a constant-temperature water bath magnetic stirring pot, and the stirring speed was 240 rpm. ICP-OES was used to determine each element’s concentration in filtered water samples. After filtration, the precipitates were dried for 12 h at 80 °C before being further examined for chemical composition, leaching stability, and morphological change. In order to ensure the reliability of the experimental results, all reactions were set up in triplicate and performed simultaneously. The removal efficiency of arsenic and leaching efficiency of zinc were calculated by Equation (1).
(1)$$\mathrm{Removal}\;\mathrm{efficiency}\;(\%)=\left[\left(V_{0}{*C}_{0}-V_{1}{*C}_{1}\right)/V_{0}{*C}_{0}\right]\times 100$$
where V_0_ (mL) is the initial volume of the wastewater, C_0_ (mg/L) is the arsenic concentration of the untreated wastewater, V_1_ (mL) is the final volume of the wastewater after treated by lead slag, and C_1_ (mg/L) is the final arsenic concentration of the treated wastewater.

Arsenic-bearing solid precipitates produced in the batch experiments were further tested for stability according to the United States Environmental Protection Agency’s Toxicity Characteristic Leaching Procedure (TCLP) test [27]. The pH value of the leaching standard solution was 2.88 ± 0. 05. It was mixed with the precipitate at a liquid–solid ratio of 20 mL/g, and shaken continuously for 18 h at 25 °C. After the reaction, the supernatant was taken and filtered with a 0.22 μm membrane filter, and the concentration of heavy metal ions in the filtrate was detected by inductively coupled plasma–optical emission spectrometer (ICP-OES, PerkinElmer Optima 5300 DV, Waltham, MA, USA).

### 2.3. Chemical Analysis of Solid Phase

XRD spectra were collected to analyze the crystalline phase of the solid samples in the experiments using an X-ray diffractometer (Advance D8, Bruker Corp, Karlsruhe, Germany) equipped with Cu-Kα radiation at a scanning rate of 5°/min in the scanning angle (2θ) range of 5–80°. The chemical and mineral properties of the precipitate were examined using a Nicolet IS 10 spectrometer (Thermo Fisher Scientific, Waltham, MA, USA) to collect FTIR spectra (KBr mode) in the range of 400–4000 cm^−1^ with a set resolution of 4 cm^−1^ and 30 scans. The morphological features and structural composition of solids were observed by scanning electron microscopy (SEM, TESCAN MIRA LMS, Brno, Czech Republic) coupled with energy-dispersive X-ray spectroscopy (EDS, Xplore). The XPS spectra were obtained using a monochromatic Al-K X-ray source (Thermo Fisher Scientific K-Alpha, Waltham, MA, USA) with a pass energy of 50 eV and a step size of 0.05 eV. The binding energy of C1s = 284.80 eV was used as the energy standard for charge adjustment. The microstructure of the precipitate was observed through a transmission electron microscope (TEM, Thermo Fisher Scientific, Talos F200X, Waltham, MA, USA) coupled with energy-dispersive X-ray spectroscopy (EDS), with an LaB6 filament, operating at 200 kV on a JEOL JEM-2100 microscope.

## 3. Results and discussion

### 3.1. Characterization of Lead Slag

The lead slag was mainly composed of wustite (Fe_0_._9536_O), calcite (CaCO_3_) gypsum (CaSO_4_.H_2_O), and magnetite (Fe_3_O_4_), as shown in the XRD pattern (Figure 1). The chemical composition of the lead slag is showed in Table 1; the content of iron and silicon is 36.93% and 12.28%, respectively, and the sum of the calcium and magnesium content is about 12%. A paste pH experiment [28] of lead slag was conducted in deionized water (DI) at a 1:2.5 g/mL solid–liquid ratio and stirred for an hour, and the pH value of leachate was 7.82, indicating that alkaline oxides in the lead slag may have a neutralizing effect on the wastewater and facilitate the removal of arsenic.

### 3.2. Thermodynamic Analysis

To forecast the dissolution behavior of lead slag and the precipitation of iron arsenate in acidic arsenic-containing wastewater, the electrochemical potential versus pH (Eh–pH diagram) of the Fe-As-H_2_O system was plotted using Fact Sage™. Figure 2 shows the stability fields for the Fe-As-H_2_O system at 25 °C and atmospheric pressure. According to thermodynamic calculations, pH levels have a major impact on the types of iron and arsenic compounds. Iron and arsenic generate iron arsenate at pH 0–8 with excess iron in the form of Fe(III), and excess iron in the form of Fe(OH)_3_ at pH 0.8–4. Arsenate can be immobilized by Fe(OH)_3_ by adsorption when the pH is greater than 4.3, when iron and arsenic no longer precipitate as iron arsenate. The valence state of iron and arsenic is similarly influenced by the redox potential. Arsenic exists in the form of As_2_O_3_ molecules in a strongly acidic environment with a low redox potential [29], whereas iron exists as divalent ions. Neither electrostatic adsorption nor the generation of iron arsenate precipitation make it easy to remove arsenic in this condition. As the redox potential and pH increase, arsenic is gradually oxidized and ionized, enabling the precipitation of iron arsenate or scorodite. To remove arsenic in the first stage, it is important to elevate redox potential and keep the initial solution pH below 4.3 by generating FeAsO_4_ precipitate, and then to remove arsenic thoroughly in the second stage by Fe(OH)_3_ adsorption [30].

### 3.3. Removal of Arsenic from Wastewater

#### 3.3.1. Effect of Dosage of H_2_O_2_

The source of arsenic in smelting wastewater is As_2_O_3_-bearing flue dust from smelting operations; thus, trivalent arsenic is predominant, with a ratio of As(III)/As (total) equal to 67% in lead smelting wastewater [7,31]. The significance of pre-oxidation is that the mobility and toxicity of As(III) are much greater than those of As(V) [32]. It was demonstrated in the last section that As(III) exists mainly in molecular form in strongly acidic environment, so it is difficult for iron oxides to adsorb it by electrostatic interaction. As the pH increases, arsenate molecules ionize and adsorb onto the iron oxide surface; however, the Langmuir constant KL values were higher for As(V) than for As(III), indicating that As(V) formed stronger bonds with Fe(III) than did As(III) [33]. However, it is difficult to transform trivalent arsenic to pentavalent arsenic under atmospheric pressure, and an oxidant needs to be introduced to accelerate the oxidation process. Previous study has proven that H_2_O_2_ dosage affects iron arsenate precipitate and controls the oxidization rate of Fe(II) and As(III) for scorodite synthesis and crystal growth [34]; therefore, in this study, we chose H_2_O_2_ as an oxidant to promote the oxidation of As(III) in wastewater.

The effect of the dosage of H_2_O_2_ on the arsenic removal from wastewater was studied at H_2_O_2_/wastewater volume ratios ranging from 0 to 10% for 12 h reaction in room-temperature batch experiments, with a solid–liquid ratio of 1 g/5 mL. As shown in Figure 3a, when the H_2_O_2_/wastewater volume ratio was 10%, the residual arsenic concentration in wastewater decreased sharply from an initial value of 6500 mg/L to 0.59 mg/L, corresponding to an arsenic removal efficiency of nearly 100%. When there was no addition of H_2_O_2_, the removal efficiency of arsenic was only 34.9%, and the residual arsenic concentration in the wastewater was 4035 mg/L. The results showed that higher H_2_O_2_ dosage leads to higher arsenic removal efficiency; this might imply that the pathway of arsenic precipitate was related to the valence of arsenic, with As(V) forming precipitate more readily with iron or adsorbing to iron oxide surfaces for removal. With the increasing of H_2_O_2_ dosage, the concentration of residual iron decreased (Figure 3c), and the toxicity of the precipitates in TCLP test decreased (Figure 3b); the reason may be that the increase of H_2_O_2_ dosage led to the oxidation of As(III) and Fe(II), promoting the generation of iron arsenate precipitate, while the hydroxide formed by the hydrolysis of Fe(III) had a strong adsorption capacity for arsenic [35]. The higher dosage of H_2_O_2_ promoted the oxidation of Fe(II), and the Fe(III) generated was hydrolyzed; the acidity generated by hydrolysis then led to a decrease in wastewater pH [36].

#### 3.3.2. Effect of the Solid–Liquid Ratio

Lead slag is the iron source for arsenic removal by iron arsenate precipitation; it will be dissolved in the acidic wastewater, and its dosage will directly affect the pH value of the wastewater and concentrations of Fe and Ca. The undissolved particles will act as nucleation sites for gypsum and scorodite, so the dosage of lead slag may also affect the size and crystallinity of the precipitate [19]. A batch experiment at 25 °C under atmospheric pressure was conducted to explore the influence of the dosage of lead slag on arsenic removal efficiency and leaching stability. The dosage of lead slag was controlled by adjusting the solid–liquid ratio to 1:3, 1:5, 1:7, 1:10, and 1:15 g/mL. Figure 4a shows that the residual concentration of arsenic sharply decreased from 2931.54 to 3.35 mg/L. When the solid–liquid ratio raised from 1:15 to 1:10 g/mL, removal efficiency increased from 56.57% to 92.73%. As a result, after the solid–liquid ratio reached 1:10, the removal efficiency no longer changed drastically, and the leaching toxicity of arsenic-containing precipitate also dropped to below 5 mg/L (Figure 4b), lower than national standards for leaching toxicity [37]. Notably, the leaching toxicity of As-bearing precipitate in the TCLP test decreased significantly with the increase of the solid–liquid ratio; when the solid–liquid ratio was 1:15, the leaching toxicity was 13.24 mg/L, and when the solid–liquid ratio increased to 1:10, the leaching toxicity was 3.36 mg/L, which meets the threshold value of designating hazardous waste. The concentration of residual iron ions in the wastewater showed an opposite trend with an increasing solid–liquid ratio, as seen in Figure 4c. The acidity of wastewater can leach limited iron ions, while—with the increase of the solid–liquid ratio—the alkalinity of lead slag can raise the pH of wastewater and promote the hydrolysis of iron ions. Considering the economic cost and environmental safety factors, we believe that a solid–liquid ratio of 1:10 is the best condition for arsenic removal and stabilization from wastewater by lead slag.

#### 3.3.3. Effect of Reaction Time

Under room temperature conditions, the effect of reaction time on wastewater component and precipitate properties was explored with a solid–liquid ratio of 1:10 g/mL and an H_2_O_2_ volume ratio of 10%. The arsenic concentration decreased drastically within 2 h (Figure 5a), which was related to the significant increase in pH of the wastewater as shown in Figure 5b. Correspondingly, the leaching concentrations of Ca, Mg, and Al increased steeply within 2 h, as shown in Figure 5c, and iron followed the same trend. One possible theory is that the iron oxides and alkaline oxides in lead slag dissolved under the acid corrosion of sulfuric acid in wastewater, releasing a large amount of iron, calcium, and magnesium ions and raising the pH, which promoted the formation of poorly crystalline iron arsenate or other co-precipitates associated with Ca-Mg-Al in weakly acidic solutions [38]. It has been demonstrated that in acidic arsenic-containing solutions, the association of Ca(II)-Fe(III)-As(V) caused the precipitation of poorly crystalline arsenic precipitate or co-precipitation products from As(V)-Fe(III) solutions, which were then adsorbed on the surface of hydrated iron oxide through surface coordination. The reaction can be split into three periods based on the trend of decreasing arsenic concentration in wastewater: (1) quick period, (2) slow period, and (3) plateau period.

The quick period of reaction was 0–2 h, during which the dissolution of alkaline oxide raised the pH of wastewater to 2.54, and a large amount of iron, calcium, and magnesium ions promoted the formation of iron arsenate precipitate and co-precipitate, lowering the concentration of arsenic to 1000 mg/L, which can be described with the following reactions:$$H_{3}AsO_{3}+H_{2}O_{2}\to H_{3}AsO_{4}+H_{2}O$$
$$FeO+Fe_{3}O_{4}+H_{2}O_{2}+6H_{2}SO_{4}\to 2Fe_{2}{\left(SO_{4}\right)}_{3}+7H_{2}O$$
$$2H_{3}AsO_{4}+Fe_{2}{\left(SO_{4}\right)}_{3}\to FeAsO_{4}+3H_{2}SO_{4}$$
$$CaCO_{3}+H_{2}SO_{4}+H_{2}O\to CaSO_{4}.2H_{2}O+CO_{2}$$

According to previous research, not all arsenic in wastewater can be eliminated by precipitation and co-precipitation; in reality, part of arsenic will be adsorbed on the surface of ferrihydrite and progressively changed into poorly crystalline iron arsenate precipitate [39,40,41]. The transformation progression can be described with the following reactions [42]:$$\equiv FeOOH_{\left(S\right)}+H_{3}AsO_{4}\to \equiv FeH_{2}AsO_{4\left(ads\right)}+H_{2}O$$
$$\equiv FeH_{2}AsO_{4(ads)}+\left(2+x\right)H_{2}O\to \equiv FeAsO_{4}.\;\left(2+x\right)H_{2}O_{\left(p-c-surfprec\right)}+2H^{+}$$
$$\equiv FeAsO_{4}.\;\left(2+x\right)H_{2}O_{\left(p-c-surfprec\right)}\to FeAsO_{4}.\;2H_{2}O\left(crys\right)+H_{2}O$$
where (≡) represents the ferrihydrite surface; (*ads*) represents adsorption; (p-c-surfprec) represents poorly crystalline surface precipitate; (*crys*) represents crystalline. After 12 h of reaction, the concentration of arsenic in the wastewater decreased to 3.35 ± 1.94 mg/L, and the removal efficiency reached 99.95%. As the reaction proceeded, the stability of the precipitate increased gradually, and the leaching toxicity of arsenic decreased from 28.02 at 1 h to 3.36 at 12 h.

From the XRD results (Figure 5d), the main components of the precipitate were gypsum and a small amount of undissolved iron oxide; the source of newly formed gypsum was calcite in lead slag, which dissolves in acid wastewater and then combines with sulfate. No diffraction peaks of scorodite or iron arsenate were detected in XRD analysis, probably because the low crystallinity of the iron arsenate precipitate or the weaker signal of amorphous iron arsenate were shielded by gypsum. These findings are consistent with the results of Duan and Li [26,43]. The difference is that the diffraction peaks of scorodite were observed in the precipitation of the reaction for 12 h by Li et al., indicating that the formation of scorodite crystals takes some time; however, in our study, the formation of scorodite was not observed, which may be caused by the different reaction conditions.

#### 3.3.4. Effect of Reaction Temperature

Reaction temperature affects the rate of nucleation and crystal growth [44], and the scorodite synthesized at a high reaction temperature has larger particle size [21]. Although studies have demonstrated that scorodite crystals grow better under high temperature and pressure settings [45], there is experimental evidence that it can be formed around 30 °C [21,46]. In this studythe precipitate obtained at 85 °C had a higher crystallinity than the one obtained at 25 °C according to SEM analysis (Figure 6). A small number of fusiform crystals can be seen on the surface of the precipitate obtained at 85 °C; this is probably scorodite or other iron arsenate crystalline matter. As a result, precipitate with high crystallinity was more safe, with arsenic leaching toxicity of 0.61 mg/L (Figure 7b), which was lower than precipitates obtained at 25 °C (3.36 mg/L).

As shown in Figure 7c, a high temperature promoted the hydrolysis of iron ions, resulting in a decrease in the concentration of residual iron in the wastewater with increasing temperature. However, no scorodite diffraction peaks were found in the precipitates collected after 12 h of reaction at 25–85 °C Nevertheless, the removal efficiency of arsenic was hardly affected by temperature, and a very ideal arsenic removal effect could be achieved at room temperature (Figure 7a). Therefore, we believe that ambient conditions are sufficient for the efficient removal and stabilization of arsenic from wastewater.

#### 3.3.5. Arsenic Removal Mechanism by Lead Slag

The FTIR spectra were used to evaluate the surface properties of arsenic-bearing precipitates obtained at different times (Figure 8). The peaks at 3543 and 3402 cm^−1^ can be assigned to the O-H stretching vibrations of the two water molecules of gypsum [39,47], whereas the O–H bending vibration from the water encapsulated in gypsum or iron arsenate is responsible for the peak at 1622 cm^−1^. The band at 827 cm^−1^ is ascribed to the As–O stretching vibration of the As–O–Fe coordination of iron arsenate precipitate [39,48,49]. As the reaction proceeded, the band width gradually decreased, indicating an increase of crystallinity. The As–O–Fe bidentate–binuclear coordinating to ferrihydrite is attributed to this band, demonstrating arsenate adsorption on ferrihydrite [43]. A broad peak appeared at 1136 cm^−^^1^ is assigned to structural SO_4_^2-^ ions coordinating with CaSO4.2H_2_O or ferric sulfate compounds [50,51,52]. Additionally, the peak at 673 cm^−1^ is incorporated into the forming of CaSO_4_.2H_2_O or poorly crystalline iron arsenate by substitution of AsO_4_^3-^ ions during synthesis from sulfate medium [14,53,54]. The peak at 607 cm^−1^ corresponds to Fe-O derived from undissolved Fe_3_O_4_ [55]. Another peak at 474 cm^−1^ is probably attributed to the Si–O asymmetric stretching vibration coordinating with the newly formed silicates [6,51].

In order to gain further insight into the mechanism of arsenic removal from arsenic-containing wastewater by lead slag, XPS scanning were performed on lead slag and arsenic-bearing precipitates obtained at 25 °C and 85 °C (Figure 9). The evolution of the valence state of precipitates was further demonstrated by the XPS narrow-scan spectra of Fe (2p_3/2_) and As (3d_5/2_), illustrated in Figure 9, and their corresponding peak fitting parameters were summarized in Table 3, Table 4 and Table 5.

As shown in Figure 9a, due to the presence of small amounts of arsenic, the characteristic peak of arsenic was also detected in the lead slag. However, compared with the As3d peak of lead slag, the As3d peaks of As-bearing precipitates obviously shifted to higher energy levels, indicating that arsenic in the wastewater was oxidized and precipitated in the form of As(V). Peaks at 43.18, 43.7, 44.31, and 44.8 (Fe-As-O) [20,39], as well as 45.39 and 46.16 eV (As-O-OH) [56], were observed, indicating the formation of iron arsenate or scorodite and that a portion of arsenic is adsorbed on the iron hydroxide surface.

The spectrum of Fe (2p_3/2_) shown in Figure 9b indicates that the precipitate’s peak was clearly shifted to a higher energy level compared to the lead slag. The percentage of Fe(II)-O in fresh lead slag was 18.58%, and the Fe(II) could effectively control the supersaturation of iron during dissolution, while no Fe(II)-O was detected in the precipitate, indicating that it had been oxidized. The peaks at 710.07, 710.85, 711.73, 712.86, and 713.32eV represent scorodite, iron-oxyhydroxide, hematite, and jarosite, respectively [20,57,58]. Fe-As(V)-O in the Fe2p peak corresponds to Fe-As(V)-O in the As3d peak, further demonstrating the formation of iron arsenate or scorodite precipitates, which agrees with the FTIR results.

The morphology and elemental composition of the precipitates generated at different reaction times were examined using SEM-EDS, as shown in Figure 10 and Table 6. The lead slag particles become smaller after 1 h of reaction due to the dissolving impact of acidic wastewater, and the previously relatively flat surface (Figure 10a,b) was wrapped in fine strips and irregular fine particles, as shown in Figure 10c,d. As the reaction continued for 6 h, the precipitate amount increased, and sediment adherence caused the surface to become denser. According to EDS data, the strips were primarily made up of Ca, S, and O, most likely gypsum phase, which also agrees with the XRD findings (Figure 5d). The gypsum observed in SEM by Li et al. had a similar striped structure. Most of the microscopic particles adhered to the surface of gypsum were composed of Fe, As, and O, and were most likely co-precipitate-doped amorphous iron arsenate; no diffraction peak was observed in XRD because of the low crystallinity. The precipitate contained fine granular iron arsenate and streaks or plates of gypsum, as shown by the EDS results in Figure 10g,i, as well as agglomerated flocs adsorbed on its surface that may represent ferric hydroxide or silica gel created by the dissolution of silica oxides. After 12 h of reaction, the amount of arsenic fixed in precipitate increased noticeably from 3.45 to 7.91 at wt.%, showing that the second half of the reaction is equally crucial for the removal of arsenic.

Transmission electron microscopy was used to examine the internal structure of the arsenic-containing precipitate formed after 12 h of reaction at room temperature, as shown in Figure 11. The precipitate’s exterior was encapsulated in dispersed, erratic gelatinous material, while its inside was composed of more regularly formed particles. The results of the energy spectrum show that the inner particles are iron arsenate, and the outer part may be iron hydroxide and silica gel generated by the dissolution of silicon oxide. Due to the semi-encapsulation and adsorption of surface colloids, the precipitation performed well in the toxin leaching process, reducing the risk of secondary contamination.

Based on the experimental data and characterization results, we propose a reaction mechanism of arsenic removal and stabilization from wastewater by lead slag, as shown in Figure 12. Lead slag dissolved in acidic arsenic-containing wastewater, and alkaline oxide dissolution raised the pH value of the wastewater and provided Ca^2+^, Fe^2+^, Fe^3+^, Mg^2+^, Al^3+^, and SiO_3_^2−^. Ca^2+^ first combined with sulfate in the wastewater to form CaSO_4_.2H_2_O; Fe^2+^ and As^3+^ were oxidized by H_2_O_2_, combined to form iron arsenate, and gradually transformed to the scorodite. Gypsum and iron arsenate used undissolved lead slag particles as growth sites. Notably, the precipitate’s surface was covered with iron hydroxide and silicate colloid formed by hydrolysis; semi-encapsulation and adsorption further enhanced the leaching stability of the precipitate.

#### 3.3.6. Prospects of Treating Wastewater Using Lead Slag

Our method calls for a low temperature, and good arsenic removal efficiency can be achieved at room temperature. The room-temperature precipitate performed well in leaching toxicity tests, and fulfills the concentration limit of less than 5 mg/L despite having little crystallinity and no scorodite production. The environmental safety of lead slag as a wastewater treatment agent is also reliable; as shown in Table 7, after 12 h of reaction, the presence of lead was no longer detectable in the wastewater, and the Pb leaching concentration of the precipitate was less than 5 mg/L limit in “Identification Standard for Identification of Hazardous Wastes” (GB 5085.3-2007), as shown in Table 8.

If lead slag can be used to treat acidic arsenic-containing wastewater, the disposal of slag and wastewater in the smelter can be solved at the same time, and the wastewater treatment cost and raw material transportation cost will be greatly reduced. This method will provide a reference value for the waste-disposal problem of non-ferrous metal smelting, and for the sake of understanding, a prospective process was designed as shown in Figure 13.

## 4. Conclusions

In this work, lead slag was used as in situ iron source and neutralizer to eliminated arsenic in the form of iron arsenate and arsenic-doped gypsum. The dissolution of lead provides alkalinity to neutralize the acidity of the wastewater, and releases large amount of Ca, Fe, Al, Mg, and Si ions. As(III) and Fe(II) was oxidized by H_2_O_2_, and the arsenic was mainly eliminated in the form of low-crystalline iron arsenate. In addition, there was a large amount of strip gypsum in the precipitate, which immobilized part of the arsenic by co-precipitation during the formation process and provided attachment sites for the iron arsenate. A 99.95% arsenic removal efficiency was achieved with a solid–liquid ratio of 1:10 g/mL, a 10% H_2_O_2_ volume ratio, and a 12 h reaction duration at room temperature. Semi-encapsulation of As-bearing precipitate by silicate colloid and ferric hydroxide enhanced the leaching stability, with As concentration of 3.36 mg/L and Pb concentration of 2.93 mg/L, which are lower than the leaching threshold (5 mg/L) in “Identification Standard for Identification of Hazardous Wastes” (GB 5085.3-2007). Lead slag has great potential in the treatment of acidic arsenic-containing wastewater and can form a joint slag–wastewater treatment and water recycling system in the smelter, paving a new direction for non-ferrous metal smelting wastewater treatment.

## Figures and Tables

**Figure 1 materials-15-07471-f001:**
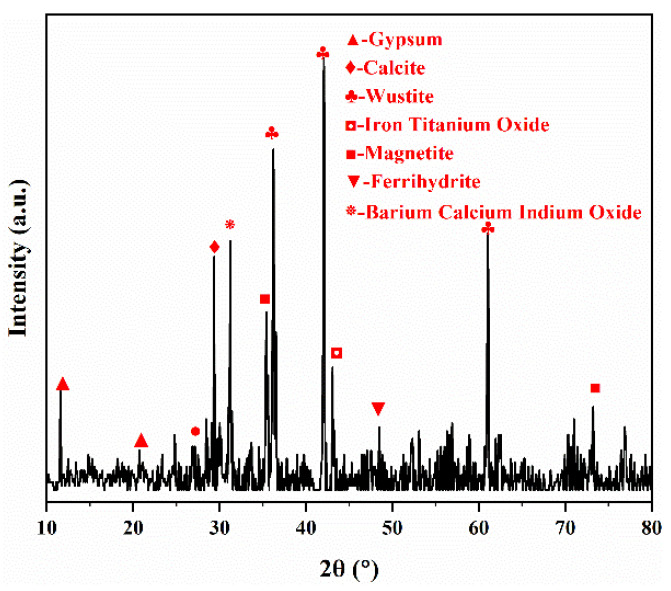
The XRD pattern of the lead slag.

**Figure 2 materials-15-07471-f002:**
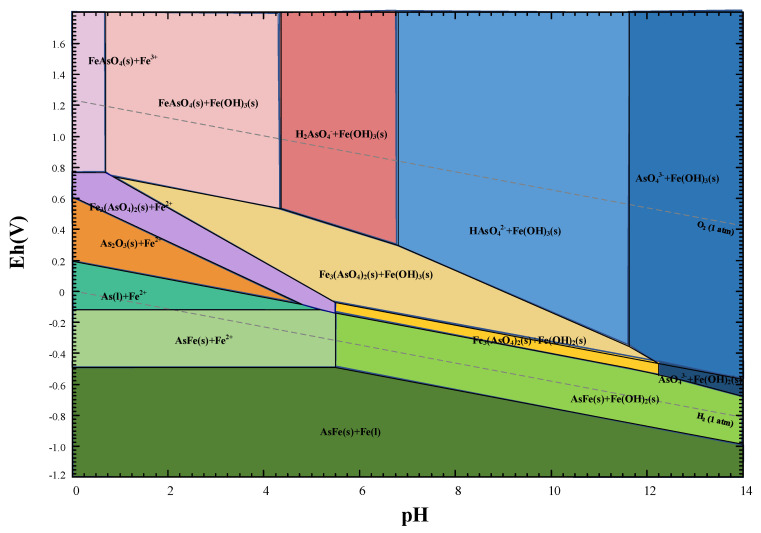
Eh–pH diagram for the As-Fe-H_2_O system at 25 °C and atmospheric pressure.

**Figure 3 materials-15-07471-f003:**
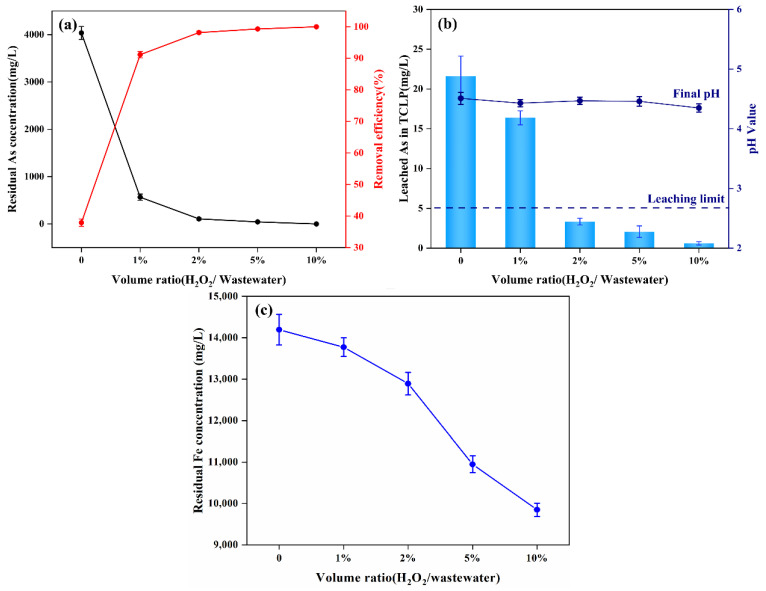
Variations of residual concentration and removal efficiency of As (**a**), leached arsenic concentration in TCLP and final pH value of wastewater (**b**), residual concentration of Fe (**c**) in bath experiments at volume ratio (H_2_O_2_/wastewater) of 0%, 1%, 2%, 5%, and 10%.

**Figure 4 materials-15-07471-f004:**
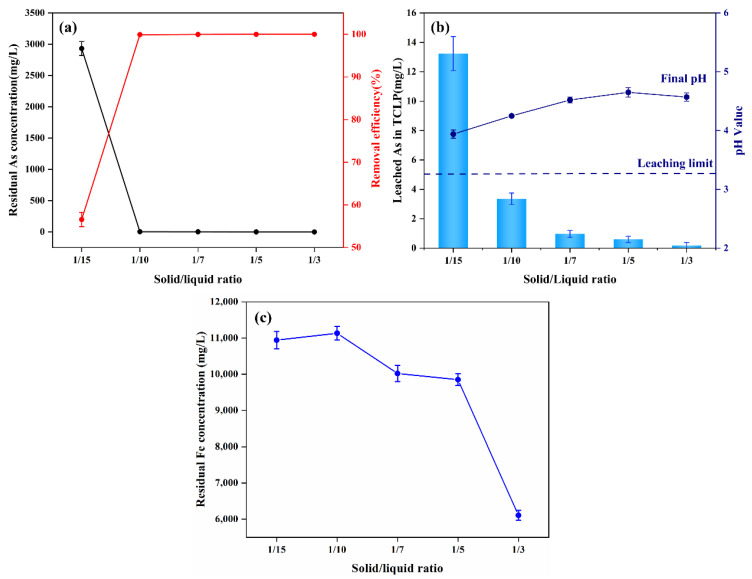
Variations of residual concentration and removal efficiency of As (**a**), leached arsenic concentration in TCLP and final pH value of wastewater (**b**), residual concentration of Fe (**c**) in bath experiments at solid–liquid ratio of 1:15, 1:10, 1:7, 1:5, and 1:3.

**Figure 5 materials-15-07471-f005:**
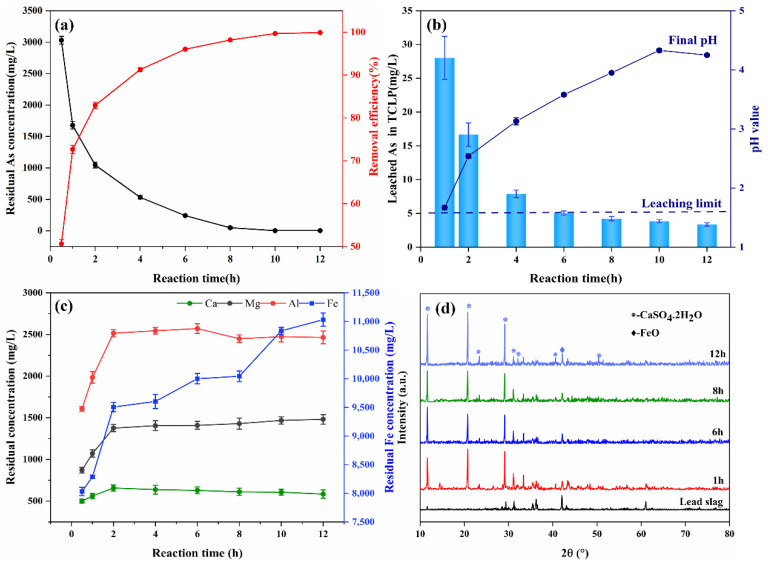
Variations of residual concentration and removal efficiency of As (**a**), leached arsenic concentration in TCLP and final pH value of wastewater (**b**), Variations of concentration of Ca, Mg, Al, Fe (**c**), XRD pattern of the arsenic-loaded precipitates obtained from bath experiments at reaction time of 1, 6, 8, and 12 h (solid–liquid ratio of 1:10) and lead slag (**d**).

**Figure 6 materials-15-07471-f006:**
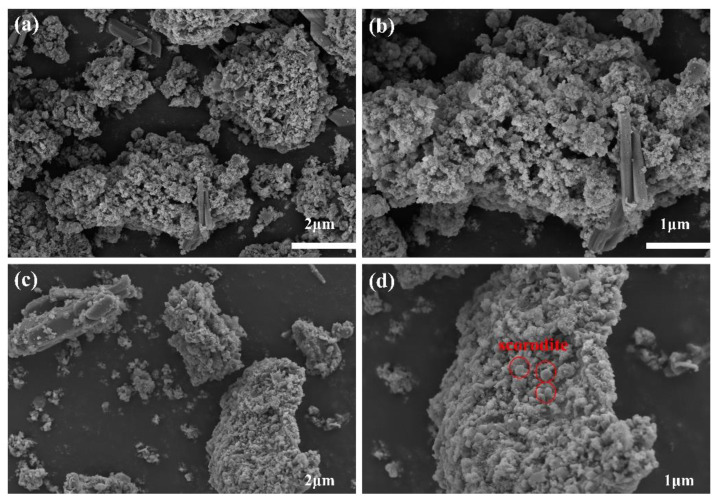
SEM images of As-bearing precipitate obtained at 25 °C (**a**,**b**), As-bearing precipitate obtained at 85 °C with a solid–liquid ratio of 1:10 and H_2_O_2_ volume ratio of 10% (**c**,**d**).

**Figure 7 materials-15-07471-f007:**
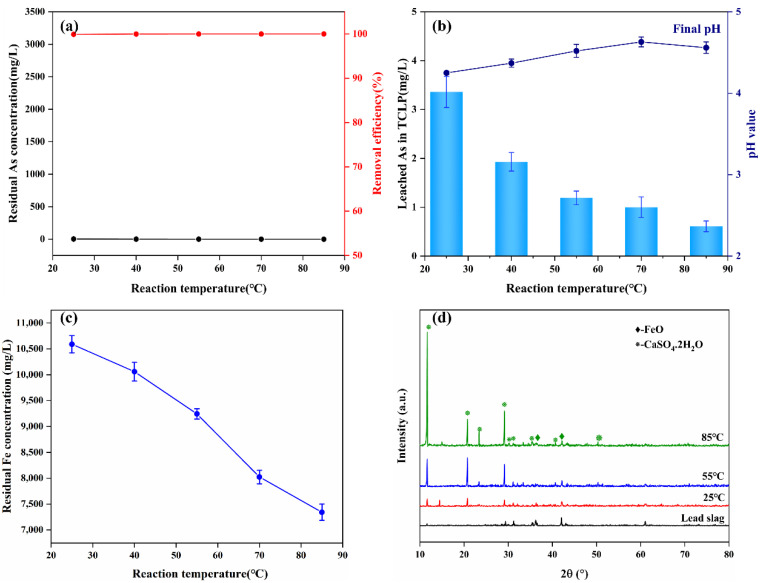
Variations of residual concentration and removal efficiency of As (**a**), leached arsenic concentration in TCLP and final pH value of wastewater (**b**), residual concentration of Fe in bath experiments at reaction temperature of 25, 40, 55, 70, and 85 °C (**c**), XRD pattern of the arsenic-loaded precipitates obtained from bath experiments at reaction temperature of 25, 55, and 85 °C (solid–liquid ratio of 1:10 and H_2_O_2_ volume ratio of 10%) and lead slag (**d**).

**Figure 8 materials-15-07471-f008:**
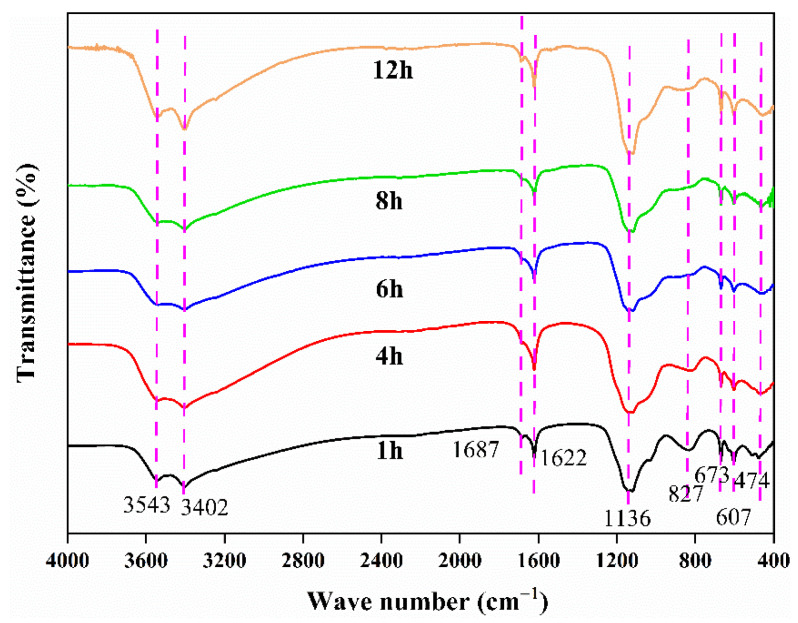
FTIR spectra of the arsenic-bearing precipitates obtained at various reaction times (1,4, 6, 8, and 12 h).

**Figure 9 materials-15-07471-f009:**
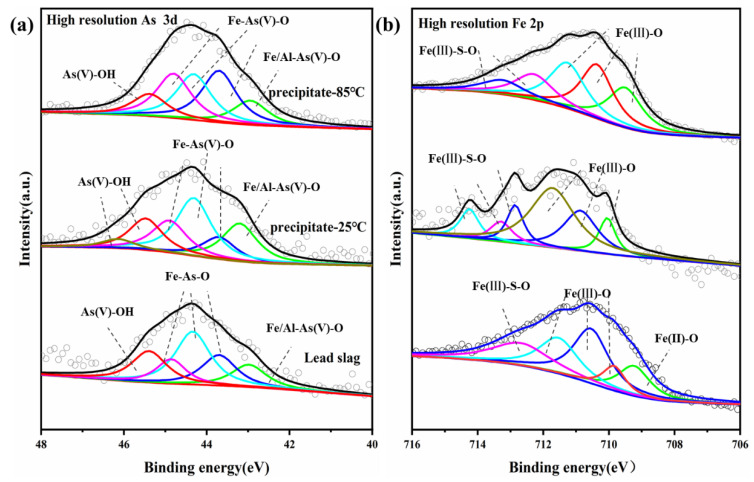
XPS spectra of (i) lead slag, As-bearing precipitates obtained after reaction at 25 °C (ii) and 85 °C (iii): (**a**) High-resolution As 3d, (**b**) High-resolution Fe 2p.

**Figure 10 materials-15-07471-f010:**
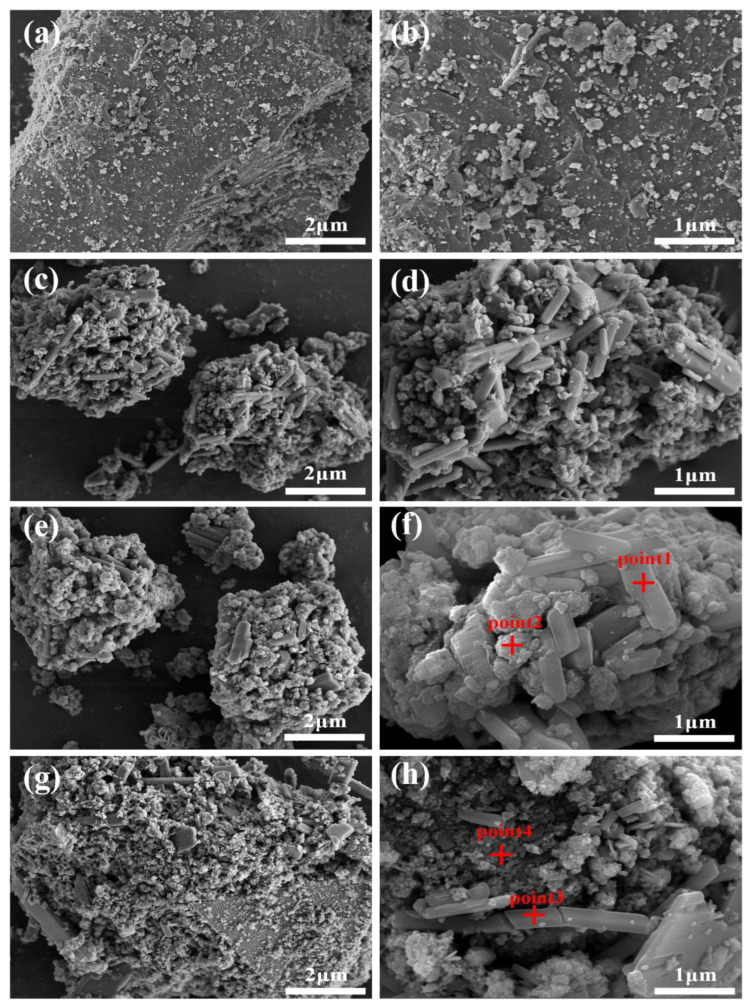
SEM-EDS images of original lead slag (**a**,**b**) and As-bearing precipitates obtained after reaction times of 1 h (**c**,**d**), 6 h (**e**,**f**), 12 h (**g**,**h**), at a 1:10 solid–liquid ratio, an H_2_O_2_ volume ratio of 10%, and a reaction temperature of 25 °C.

**Figure 11 materials-15-07471-f011:**
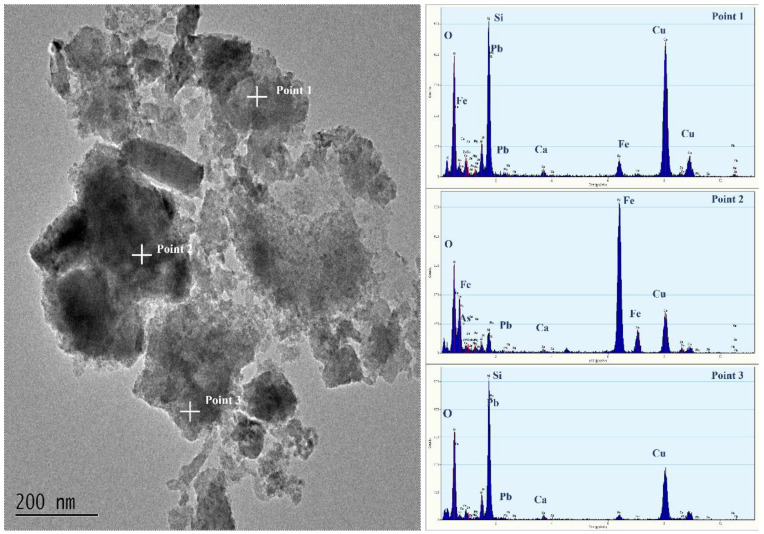
TEM-EDS images of arsenic-bearing precipitate obtained after reaction times of 12 h at a 1:10 solid–liquid ratio, an H_2_O_2_ volume ratio of 10%, and a reaction temperature of 25 °C.

**Figure 12 materials-15-07471-f012:**
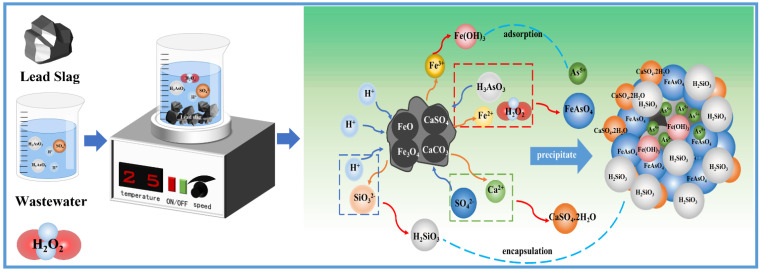
Mechanistic diagram of arsenic removal and immobilization in the treatment of arsenic-bearing wastewater using lead slag.

**Figure 13 materials-15-07471-f013:**
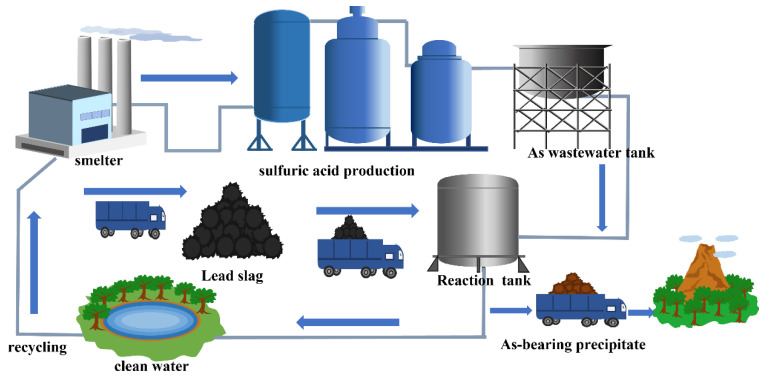
Flow diagram of co-treatment of lead smelting slag with arsenic-containing wastewater.

**Table 1 materials-15-07471-t001:** Chemical composition of lead slag (wt. %).

Elements	Concentration (wt. %)	Elements	Concentration (wt. %)	Elements	Concentration (wt. %)
O	19.60	Pb	2.71	As	0.54
Fe	36.93	S	2.62	Cu	0.36
Si	12.29	Mn	1.56	Cl	0.30
Ca	11.05	Mg	0.87	Na	0.24
Al	4.96	K	0.77	Ba	0.22
Zn	3.81	Ti	0.64	others	0.53

**Table 2 materials-15-07471-t002:** Chemical composition of the arsenic wastewater (mg/L).

Element	As	Pb	Fe	Zn	Sb	Cd	H_2_SO_4_
Concentration	6500 ± 12	4.50 ± 0.12	35.20 ± 1.04	21.50 ± 0.72	15.89 ± 0.57	87.50 ± 1.49	56,000 ± 49

**Table 3 materials-15-07471-t003:** XPS data and chemical states of the lead slag.

Spectral Peak	BE (eV)	FWHM	Area (%)	Chemical States	Comment
As3d	42.98	1	14.13%	As(III)-S	Estratite
As3d	43.7	1	19.30%	Fe-As(V)-O	Amorphous iron arsenate/Scorodite
As3d	44.34	1	33.53%	Fe-As(V)-O	Amorphous iron arsenate/Scorodite
As3d	44.86	0.85	13.12%	Fe-As(V)-O	Amorphous iron arsenate/Scorodite
As3d	45.39	1	19.92%	As-O(V)-OH	Adsorbed arsenate
Fe2p	709.24	1.33	18.58%	Fe(II)-O	Ferrous oxide
Fe2p	710.04	1.2	23.18%	Fe(III)-As-O	Amorphous iron arsenate/Scorodite
Fe2p	710.80	1.18	22.17%	Fe(III)-O	Hematite/Magnetite
Fe2p	711.70	1.48	23.41%	Fe(III)-O	Iron-oxyhydroxide
Fe2p	712.81	1.76	12.66%	Fe(III)-O	Iron-oxyhydroxide

**Table 4 materials-15-07471-t004:** XPS data and chemical states of the As-bearing precipitate obtained at 25 °C.

Spectral Peak	BE (eV)	FWHM	Area (%)	Chemical States	Comment
As3d	43.14	1.36	17.56%	Fe/Al-As(V)-O	Fe/Al-arsenate solid solution
As3d	43.77	1	11.70%	Fe-As(V)-O	Amorphous iron arsenate/Scorodite
As3d	44.12	1.18	22.30%	Fe-As(V)-O	Amorphous iron arsenate/Scorodite
As3d	44.75	1.1	14.87%	Fe-As(V)-O	Amorphous iron arsenate/Scorodite
As3d	44.99	1.46	20.14%	Fe--As(V)-O	Amorphous iron arsenate/Scorodite
As3d	45.62	1.25	13.43%	As(V)-O-OH	Adsorbed arsenate
Fe2p	710.05	0.6	9.62%	Fe(III)-As-O	Amorphous iron arsenate/Scorodite
Fe2p	710.87	1.24	22.11%	Fe(III)-O	Hematite/Magnetite
Fe2p	711.72	1.6	42.23%	Fe(III)-O	Iron-oxyhydroxide
Fe2p	712.86	0.6	10.16%	Fe(III)-As-O	Amorphous iron arsenate/Scorodite
Fe2p	713.29	0.79	7.04%	Fe(III)-S-O	Fe-sulphate complex
Fe2p	714.25	0.68	8.84%	Fe(III)-S-O	Fe-sulphate complex

**Table 5 materials-15-07471-t005:** XPS data and chemical states of the As-bearing precipitate obtained at 85 °C.

Spectral Peak	BE (eV)	FWHM	Area (%)	Chemical States	Comment
As3d	42.94	1	12.84%	Fe/Al-As(V)-O	Fe/Al-arsenate solid solution
As3d	43.7	1	27.48%	Fe-As(V)-O	Amorphous iron arsenate/Scorodite
As3d	44.31	1	24.38%	Fe-As(V)-O	Amorphous iron arsenate/Scorodite
As3d	44.80	1	23.57%	Fe-As(V)-O	Amorphous iron arsenate/Scorodite
As3d	45.38	1	11.73%	As(V)-O	Adsorbed arsenate
Fe2p	709.50	1.23	21.53%	Fe(III)-As-O	Amorphous iron arsenate/Scorodite
Fe2p	710.37	1.17	27.52%	Fe(III)-O	Hematite/Magnetite
Fe2p	711.26	1.3	25.72%	Fe(III)-O	Iron-oxyhydroxide
Fe2p	712.28	1.4	15.61%	Fe(III)-As-O	Amorphous iron arsenate/Scorodite
Fe2p	713.25	1.57	9.64%	Fe(III)-S-O	Fe-sulphate complex

**Table 6 materials-15-07471-t006:** The elemental compositions (at. %) for each EDS point.

Point	O	Mg	Al	Si	S	Ca	Fe	Zn	As
1	56.84	0.21	1.49	5.51	13.64	13.01	7.33	0.51	1.47
2	59.71	1.36	2.33	8.33	3.71	1.01	17.84	2.25	3.45
3	50.38	0.15	1.69	5.33	8.21	6.53	23.79	1	2.94
4	50.58	0.09	1.67	7.4	2.55	0.37	28.29	1.13	7.91

**Table 7 materials-15-07471-t007:** Variation of Pb concentration in wastewater with reaction time (mg/L).

Reaction Time (h)	1	2	4	6	8	10	12
Concentration (mg/L)	30.42 ± 0.87	18.17 ± 0.77	1.68 ± 0.25	0.28 ± 0.05	0.15 ± 0.03	0.001	0.001

**Table 8 materials-15-07471-t008:** The Pb leaching concentration of precipitate obtained at different times (mg/L).

Reaction Time (h)	1	2	4	6	8	10	12
Concentration (mg/L)	2.70 ± 0.32	1.55 ± 0.56	1.90 ± 0.34	2.96 ± 0.24	2.88 ± 0.13	2.81 ± 0.16	2.93 ± 0.23

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
