# Peer review of "Utilization of Lead Slag as In Situ Iron Source for Arsenic Removal by Forming Iron Arsenate"

_materials, 2022, doi:10.3390/ma15217471_

Round 1

Reviewer 1 Report

Please refer to the attached file

Author Response

 We sincerely  thank the reviewer for your careful reading and thoughtful comments, which significantly  improves the quality of this manuscript.Your comments have been carefully taken into consideration in revised manuscript.

Author Response

 We sincerely  thank the reviewers for your careful reading and thoughtful comments, which significantly  improves the quality of this manuscript. Your comments have been carefully taken into consideration in revised manuscript.

Reviewer 3 Report

The paper presents the results of a study where the wastewater containing a high concentration of arsenic (c.a. 6500 mg/L), generated in a lead smelting plant, are treated for arsenic removal with the slags generated in the same process. I have the following remarks:

The novelty of the paper, over the present state of the art, must be better highlighted;

In section 2.2 the authors must precise that they use H2O2 as the oxidant agent (see line 79). Why other oxidants (such as NaClO or chlorine gas) were not considered in the study? Why the effect of pH on the process was not studied?

Line 96: not sure that immersed is the proper word; you mean the solution (why “standard”? why did it have the pH value indicated in line 97?) used for the leaching test

Line 122-125 – at this point you cannot say the effect of pH on As removal

Section 3.2 – The first sentence of the section is not complete and, in general, I do not understand the reason why this section was placed here. This consideration is true also for the subsequent sections: you must first present your results and subsequently discuss them.

With reference to Figures 3, 4, 5, 7, I think you should report either the residual As concentration or the removal efficiency, in order not to duplicate your results. With reference to graphs “b”, I do not understand why you combined an information concerning the wastewater after treatment and the pH of the eluate generated from the precipitate solids, that makes the diagram confused. Why in Figure 5 did you report the trend of several metals and in the other figures only the trend of iron? In Figure 7 is the legend of the x-axis correct?

In general, the discussion must follow the presentation of the results not vice versa. That makes quite difficult the reading of you paper.

Finally, English is not my first language, but I think useful a polishing of language and sentence structure with the help of a professional language editor.

Author Response

 We sincerely  thank the reviewer for your careful reading and thoughtful comments, which significantly  improves the quality of this manuscript. Your comments have been carefully taken into consideration in revised manuscript.

Reviewer 4 Report

The article “Utilization of bulk solid waste as in-situ iron source for the removal of trivalent arsenic in acidic smelting wastewater” is dealing with the use of lead smelting slag as in-situ iron source and neutralizer in the wastewater treatment process for As(III) removal including optimum conditions for As precipitation. Arsenic precipitates were characterized by complementary analytical techniques and As(III) removal mechanism was studied, including its toxicity by TLCP leaching tests. 

However there are a series of issues which require more explanation and revised interpretation of obtained data.  Additionally authors should demonstrate what is the novelty of this paper with respect to a recent paper (2022) and other recent papers in literature to avoid duplication (see also major comments and questions and list of detailed questions and comments).

Major questions and comments

1. Recently an article was published (Removal of arsenic in acidic wastewater using Lead–Zinc smelting slag: From waste solid to As-stabilized mineral (2022), Yongkui Li et al., Chemosphere 301, 134736) which is dealing with the same research topic using a similar approach and the same analytical techniques. Therefore authors should demonstrate what is the novelty of this paper with respect to this paper and other recent paper in literature to avoid duplication. Please comment.

2. Lead slag contains significant amounts of lead. No discussion nor experimental data are found in this manuscript concerning the fate of lead when adding slag in this acid wastewater solutions. This information is required since author claim a sustainable solution for As (II) removal from wastewater. Possible introduction of lead in the environment will mortgage the proposed strategy for As(III) removal.  Please comment. 

3. A grammar and spelling check is required and recommended for the manuscript.

4. Figures are not found (Figure 7 which is a duplicate of Figure4) or not discussed (Fig. 4 (c)) pointing to careless  and editing of this manuscript. See detailed comments and questions.

5. Captions of Tables require revision. See detailed comments and questions.

6. Interpretation of FTIR spectra requires revision. See detailed comments and questions.

7. Certain statements require rephrasing and are not confirmed by experimental data or are in conflict. (See detailed list of comments and questions)     

8. Abstract and conclusions should be revised after re-interpretation of experimental data and taken into account questions and comments of reviewer.  

Detailed list of comments and questions

Abstract

Line 18: crystalline” instead of “crystallinity”

1. Introduction

Line 25: “Large amounts” instead of “Large amount”

2. Methods and materials  

Line 66: “to pass 75μm sieve.” Instead of “to pass 75 μm sieve.”

Line 68: Type and supplier of XRF instrument should be included.

Line 68: “to <75 μm” instead of “to<75μm”

Line 69: How calibration was performed in XRF? Please comment.

Line 70: “inductively coupled plasma-optical emission spectrometer (ICP-OES)” instead of “inductively coupled plasma-emission spectrometer (ICP-OES)”  

Line 70: Type and supplier of XRF instrument should be included.

Lines 71-74: quality (grade) and supplier of chemicals should be reported.

Line 75: number of significant digits should be limited and should be consistent in Table 1 and in lines 120-121. Experimental errors (standard deviations) should be included.

Line 76: Experimental errors (standard deviations) should be included in Table 2.

Line 84: “determined” instead of “identified”

Lines 91-93: units for parameters in Eq. (1-2) should be included.

Line 99: “a 0.22 µm membrane filter” instead of “a 0.22-micron filter membrane”

Line 100: “inductively coupled plasma-optical emission spectrometer” instead of “inductively coupled plasma optical emission spectrometer”

Line 104: “experiments” instead of “experiment”

Lines 107: FTIR mode (KBr or ATR) should be included. Please comment.

Line 110: Type and supplier of XPS instrument should be included.

Line 112: ”C1s = 284.80 eV” instead of “C1s=284.80eV”

3. Results and discussion

Line 120: are there crystalline phases of Si present since Si is one of the major elements in the slag? Please comment.

Line 124: Is there any idea on the alkalinity of the used lead slag? Please comment.

Lines 129-130: “To forecast the dissolution behavior of lead slag and the precipitation of iron arsenate in acidic arsenic-containing wastewater the electrochemical potential versus pH (Eh-pH diagram) of the Fe-As-H2O system was plotted using Fact Sage™.” instead of “To forecast the dissolution behavior of lead slag and the precipitation of iron arsenate in acidic arsenic-containing wastewater. The electrochemical potential versus pH (Eh-Ph diagram) of the Fe-As- H2O system was plotted using Fact Sage™”.

Line 134: According to Fig.2 iron arsenate with excess of Fe(III) is found up to pH12. Please comment.

Lines 135 and 136: “Fe(OH)3” instead of “Fe (OH)3

Line 153: “It has been demonstrated” instead of “It has be demonstrated”

Line 179: “The higher dosage of H2O2 promotes the oxidation of Fe (II), and the Fe (III)  generated is hydrolyzed” instead of “The higher dosage of H2O2 promotes the oxidation of Fe(II), and the Fe(II) generated by its oxidation are hydrolyzed”

Line 180: Why this decrease of pH is only observed at 10% (H2O2/wastewater)? Please comment.

Line 195: “when solid-liquid ratio raised from 1:15 to 1:10 g/mL, removal efficiency increased from 56.57% to 92.73%.” instead of “when solid-liquid ratio raised from 1:15 to 1:10 g/mL, corresponding to removal efficiency increased from 56.57% to 92.73%.”

Line 196: “the removal efficiency no longer changed drastically” instead of “the removal efficiency was no longer change drastically”

Line 201: “the leaching toxicity was 13.24 mg/L” instead of “the leaching toxicity is 13.24 mg/L”.

Lines 204-205: The slag contains 2.7 wt.% Pb. What about the environmental safety when introducing large quantities lead/slag since a solid-liquid ratio of 1:10 is proposed? Please comment.

Line 207: Fig.4 (b): Why pH suddenly decreased when increasing the solid /liquid ratio? Please comment.

Line 207: Fig.4 (c) is not discussed. Please comment.

Line 235:” by precipitation and co-precipitation” instead of “by precipitate and co-precipitate”

Lines 237-240: abbreviation “p-c-surfprec”, “p-csurfprec” and crs” are not explained. Please comment.

Line  244: According to Fig.5(b), leaching toxicity of As is higher than 20.08 mg/L. Please comment.  

Line 248: “were detected” instead of “was detected”

Line 253: labels y axis in Figures 5 (a) and (c) are not correct: As should be included.

Line 265: “based on structure of scorodite”: not clear to me, which kind of structure? Please comment. When crystallinity increased of scorodite why no XRD peaks were found? XRD diffractogram of precipitate obtained at 85°C should be included. Please comment.

Line 266: “more safely” instead of “more safety”

Line 270: Why EDS analysis was not performed for labelled spots in Fig.6 (d)? Please comments.

Lines 272-277: This statement is in conflict with lines 266-268 where it is claimed that higher temperatures promote scorodite precipitate with higher crystallinity which lower toxicity and thus more safely. Additionally there is no Figure 7(c) presenting the effect of temperature. Please comment.

Line 279: Figures 7 are a copy of Figure 4. No figures are found showing the effect of temperature. Please comment.

Line 281: “Arsenic” instead of “arsenic”.

Line 283: The peaks at 3543 and 3402 cm−1 can rather be assigned to the O-H stretching vibrations of the two water molecules of gypsum as confirmed by the O-H bending vibration at 1622 cm-1. Please comment.

Lines 286-288: In my opinion there is no shift or the band at 827 cm-1, only a decrease of band width is observed (as also for other bands) indicating an increase of crystallinity.

Line 290: The broad and intense band at 1136 cm-1 can be assigned to sulphate groups as in gypsum and in other sulphates. It will be hard to proof that K-Jarosite is present only by FTIR, and it should be at least confirmed by XRD. Please comment.  

Line 296: “silicates” instead of “H2SiO3”. What is the source of H2SiO3? Please comment.

Line 298: y axis in Fig.8 should be labelled as transmission in %. Please comment. 

Lines 300-321: It is not clear which peak positions in XPS are related with As(III) and As(VI) and Fe(II) and Fe(II) respectively. Additionally the comments in Tables 3-4 and 5 are confusing since no reference is made to the oxidation state of as and Fe. Please comment.

In Tables 3-4-5: what is absorbed arsenate??? Please comment. What is the nature of the Fe-sulphate complex??? Please comment. I have strong doubts that one can achieve such detailed reliable quantitative information and differentiation between different iron species such as ferrous oxide and iron-oxyhydroxide. Please comment. 

Lines 321-324: There is not much confirmation regarding the presence of hematite (FTIR results are referring to Fe3O4 while presence of K-jarosite cannot be confirmed by FTIR). Please comment. 

Lines 329: Table 5 has the same caption (25°C) as Table 4? Abbreviations BE (binding energy) and FWHW (full width at half maximum) are not explained in caption. Please comment.

Line 335: Two hours of reaction? Fig.10 is referring to 1h, 6h and 12h reaction time. Please comment.

Line 344: Silica gel? In Line 296 authors refer to H2SiO3? What will be the solubility of H2SiO3 around pH=4.5 and in presence of H2O2? Please comment.

Lines 345-347: Fe/As ratio in Table 6 for points 1, 2, 3 and 4 are 5.0; 5.3; 8.1 and 3.6 respectively. How can one conclude from these results that the amount of arsenic as iron arsenate is increasing? Please comment.

Lines 349-356: what is the relationship between SEM and TEM results since we are looking to a different size? Please comment.   

Lines 351: What is the effect of sample preparation on TEM results since we are dealing with an nm scale and data of only three points? Additionally it was stated “that undissolved particles will act as nucleating sites for gypsum and scorodite affecting the size and crystallinity of the precipitate”. Therefore encapsulation by iron hydroxide and silicagel seems unlikely and in conflict with previous statement and statement in line 364 indicating the formation of CaSO4.2H2O (confirmed by FTIR as a major component in the precipitate) which is co-precipitating with FeAsO4. Finally no Ca nor S is detected (indicative for the presence of CaSO4.2H2O) although present according to the SEM results (Table 6). Please comment.

Lines 364: What is the source of SiO32- which is mentioned at different occasions in this manuscript? Please comment.

Lines 371: Because of all previous comments I cannot agree with the mechanistic diagram presented in Fig.12. Especially presence of H2SiO3 and SiO32- in presence of H2O2 is not clear (and their insolubility in water except in an alkaline medium). Please comment.

4. Conclusions

Conclusions should be revised taking into account comments and questions of the reviewer as well the fate of lead as important component in the slag. Please comment.

Author Response

(The authors gave the same response as above.)

Round 2

Reviewer 2 Report

Author has not revised as previous suggestions. Please see the attachment.

Author Response

(The authors gave the same response as above.)

Reviewer 3 Report

The authors replied to the comments in a satisfactory way and improved the quality of the manuscript accordingly.

Author Response

We sincerely thank you for your careful reading and professional comments, which have helped a lot in improving the quality of the manuscript.

Reviewer 4 Report

I have read the revised version of the article “Utilization of bulk solid waste as in-situ iron source for the removal of trivalent arsenic in acidic smelting wastewater” and the comments and answers to the reviewers. Authors have addressed the comments and questions of the reviewers in a detailed way and have implemented changes and additional information.

However considering the answers on my comments and questions and the revised manuscript, some of these questions were not answered properly and/or no appropriate changes in the manuscript were made where asked. Therefore I have to repeat these comments and questions for answering these comments and implement the changes in the manuscript when necessary. Additionally new questions popped up when reading this revised version.

I hope authors will again consider these questions for a better understanding and improvement of this paper. 

I regret that in this form the manuscript is still not suitable for publication. 

Questions and comments

1. Recently an article was published (Removal of arsenic in acidic wastewater using Lead–Zinc smelting slag: From waste solid to As-stabilized mineral (2022), Yongkui Li et al., Chemosphere 301, 134736) which is dealing with the same research topic using a similar approach and the same analytical techniques. No answer was received on this comment.

Authors should include this reference in the list of references and refer to it in the introduction (Lines 51-57). Results and outcomes should be compared and discussed in the results section as a similar approach and similar analytical techniques (Lines 58-65) were used. Finally authors should demonstrate what is the novelty of this manuscript with respect to this paper to avoid duplication.

2. Line 32: It is not clear why photocatalysis is referenced as an important method for As removal by ref. 8 and adsorption is referenced by reference 9  pointing to removal of an organic dye as an important method for As removal instead of activated carbon? Please comment.

3. Line 72: Type and supplier of XRF instrument should be included. Please comment.

4. Line 81: Table 2: Editing of mean and standard deviation is not clear and should be adapted. Please comment.

5. Lines 196-199: From this statement it can be concluded ““that undissolved particles will act as nucleating sites for gypsum and scorodite affecting the size and crystallinity of the precipitate”. Therefore encapsulation by iron hydroxide seems unlikely and in conflict with previous statements and statement in line 364 indicating the formation of CaSO4.2H2O (confirmed by FTIR as a major component in the precipitate) which is co-precipitating with FeAsO4. Additionally an important issue is the kinetics of precipitation and co-precipitation which is not discussed and is essential when dealing with mechanisms. In Yongkui Li et al., Chemosphere (2022) 301, 134736 it is  stated that ” the dissolved Ca2+ ion preferentially reacted with SO4 2− ion in the form of columnar-like CaSO42H2O precipitate, providing numerous nuclear sites as in situ “seeds “for the formation of amorphous ferric arsenate. The dissolved Fe(II) and As(III) ions were oxidized to Fe(III) and As(V) ions by H2O2, and later reacted with each other to generated amorphous ferric arsenate on the surface of CaSO42H2O, and then evolved into scorodite crystals with high stability (90°C) in the form of columnar-like CaSO42H2O precipitate, providing numerous nuclear sites for the formation of amorphous ferric arsenate. Subsequently, the amorphous ferric arsenate transformed into rosettes of intergrown scorodite crystals with highly aggregated granular nanocrystallites.” Please comment.

As mentioned in comment 1 it is essential that authors should include results of this similar investigation in the discussion and should explain why different results were obtained. Please comment

6. Lines 260-266: Authors should compare these statements with other literature data as mentioned in comments 1 and 5 (Yongkui Li et al., Chemosphere (2022) 301, 134736  

Line 281: “more safely” instead of “more safety”.

7. Line 284: Caption of Figure 6 should include reaction conditions (solid/liquid ratio; H2O2 volume ratio).

8. Line 293: Caption of Figure 7 should include reaction conditions (solid/liquid ratio; H2O2 volume ratio).

9. Line 297: “Arsenic” instead of “arsenic”.

10. Line 363: Will dissolution of silica oxides (silicates) in acid only give metasilicates and no ortho silicates? Please comment.

11. Line 365: “3.45 to 7.91 at wt.%” instead of “3.45 to 7.91”

12. Lines 367-375: What is the relationship between SEM and TEM results? Ca and S are detected in SEM-EDS analysis (Table 6). However in TEM-EDS spectra (Fig. 11) no Ca nor S is detected (indicative for the presence of CaSO4.2H2O), only silicates (Si and O and Fe-oxides (Fe and O)) are detected. Please comment.

13. Lines 371-388: As mentioned in previous comments and recent literature data (Yongkui Li et al., Chemosphere (2022) 301, 134736) no encapsulation of FeAsO4 is observed. On  the contrary, the dissolved Fe(II) and As(III) ions were oxidized to Fe(III) and As(V) ions by H2O2, and later reacted with each other to generated amorphous ferric arsenate on the surface of previously formed columnar-like CaSO42H2O. Therefore I cannot agree with the mechanistic diagram presented in Fig.12. A discussion and comparison is highly needed. Please comment.

14. Line 401: In Table 7 ND should be replaced by the detection limit. Please comment.

4. Conclusions

Conclusions should be revised taking into account comments and questions of the reviewer. Please comment.

Author Response

(The authors gave the same response as above.)

Round 3

Reviewer 2 Report

Accepted in present form

Author Response

(The authors gave the same response as above.)

Reviewer 4 Report

I have read the revised version of the article “Utilization of lead slag as in-situ iron source for arsenic removal by forming iron arsenate” and the comments and answers to the reviewers.

Authors have addressed the comments and questions of the reviewers in a detailed way and have implemented changes and additional information where needed.

I am convinced that the overall quality of the article has improved considerably and I support publication of the revised version of the article.

There are some minor spelling errors which should be corrected prior to publication.

Line 14: “have been proved to be good arsenic fixing minerals” instead of “have been proved to be a good arsenic fixing minerals”.

Line 22: “6500 mg/L” instead of “6500mg/L”

Line 24: “was 3.36 mg/L (As) and 2.93 mg/L (Pb), lower than the leaching threshold (5mg/L” instead of “was 3.36mg/L (As) and 2.93mg/L (Pb), lower than the leaching threshold (5mg/L”.

Line 64: “7530 mg/L” instead of “7530mg/L”.

Line 66: “6500 mg/L” instead of “6500mg/L”. 

Line 67: “3.36 mg/L and Pb concentration of 2.93 mg/L” instead of “3.36mg/L and Pb concentration of 2.93mg/L”.

Mine

Line 86: “to < 75μm” instead of “to<75μm”.

Line 88: “inductively coupled plasma-optical emission spectrometer (ICP-OES)” instead of “inductively coupled plasma-emission spectrometer (ICP-OES)”.

Line 127: “spectrometer” instead of “spectrometers”.

Line 219: “the removal efficiency” instead of “he removal efficiency”.

Line 316: “arsenic-bearing” instead of “Arsenic-bearing”.

Line 320: “the peak at 1622 cm-1. The band at 827 cm-1” instead of “the peak at 1622cm-1. The band at 827cm-1”.

Line 325: “at 1136 cm-1“ instead of “at 1136cm-1”.

Line 371: “after 1 hour” instead of “after 1 hours”.

Line 418: “5mg /L” instead of “5mg/L”.

Line 440: “12 hour reaction duration” instead of “12hour reaction duration”.

Line 442: “3.36 mg/L and Pb concentration of 2.93 mg/L, which are lower than the leaching threshold (5 mg/L) “ instead of “3.36mg/L and Pb concentration of 2.93mg/L, which are lower than the leaching threshold (5mg/L)”.
